# Topology stabilized fluctuations in a magnetic nodal semimetal

Nathan C. Drucker [1,2,16] ✉, Thanh Nguyen[1,3,16], Fei Han [1,3,16], Phum Siriviboon[1,4,16], Xi Luo[5,16], Nina Andrejevic[6], Ziming Zhu[7], Grigory Bednik[3], Quynh T. Nguyen[4], Zhantao Chen[8], Linh K. Nguyen[4], Tongtong Liu[4], Travis J. Williams [9], Matthew B. Stone [9], Alexander I. Kolesnikov [9], Songxue Chi [9], Jaime Fernandez-Baca [9], Christie S. Nelson [10], Ahmet Alatas [11], Tom Hogan [12], Alexander A. Puretzky [13], Shengxi Huang [14], Yue Yu[15] ✉ & Mingda Li [1,3] ✉

The interplay between magnetism and electronic band topology enriches topological phases and has promising applications. However, the role of topology in magnetic fluctuations has been elusive. Here, we report evidence for topology stabilized magnetism above the magnetic transition temperature in magnetic Weyl semimetal candidate CeAlGe. Electrical transport, thermal transport, resonant elastic X-ray scattering, and dilatometry consistently indicate the presence of locally correlated magnetism within a narrow temperature window well above the thermodynamic magnetic transition temperature. The wavevector of this short-range order is consistent with the nesting condition of topological Weyl nodes, suggesting that it arises from the interaction between magnetic fluctuations and the emergent Weyl fermions. Effective field theory shows that this topology stabilized order is wavevector dependent and can be stabilized when the interband Weyl fermion scattering is dominant. Our work highlights the role of electronic band topology in stabilizing magnetic order even in the classically disordered regime.

Topological Weyl semimetals (WSMs) host topologically non-trivial electronic bandstructures, where the band touching points carry nontrivial Berry curvature and the low-lying excitations are described by the Weyl equation[1]. Magnetic WSMs have been the subject of intense scrutiny because of their promising applications in microelectronic[2–4], optoelectronic[5,6], thermoelectric[7–10], catalytic[11], and spintronic[12,13] devices. While symmetry breaking provided by magnetism in WSMs leads to states of scientific and technological significance, it is also important to consider the impact of symmetry restoring magnetic fluctuations on their macroscopic properties.

In magnetic WSMs, thermal fluctuations near the magnetic transition temperature can lead to topological states within locally correlated fluctuating domains[14–16]. Conversely, topological Weyl fermions are present at all temperatures and potentially influence the magnetism.

[1]Quantum Measurement Group, MIT, Cambridge, MA, USA. [2]John A. Paulson School of Engineering and Applied Sciences, Harvard University, Cambridge, MA, USA. [3]Department of Nuclear Science and Engineering, MIT, Cambridge, MA, USA. [4]Department of Physics, MIT, Cambridge, MA, USA. [5]College of Science, University of Shanghai for Science and Technology, Shanghai, China. [6]Argonne National Laboratory, Lemont, IL, USA. [7]School of Physics and Electronics, Hunan Normal University, Changsha, China. [8]SLAC National Accelerator Laboratory, Menlo Park, CA, USA. [9]Neutron Scattering Division, Oak Ridge National Laboratory, Oak Ridge, TN, USA. [10]National Synchrotron Light Source II, Brookhaven National Laboratory, Upton, NY, USA. [11]Advanced Photon Source, Argonne National Laboratory, Lemont, IL, USA. [12]Quantum Design, Inc., San Diego, CA, USA. [13]Center for Nanophase Materials Sciences, Oak Ridge National Laboratory, Oak Ridge, TN, USA. [14]Department of Electrical Engineering, Rice University, Houston, TX, USA. [15]Department of Physics and State Key Laboratory of Surface Physics, Fudan University, Shanghai, China. [16]These authors contributed equally: Nathan C. Drucker, Thanh Nguyen, Fei Han, Phum Siriviboon, Xi Luo. ✉e-mail: ndrucker@g.harvard.edu; yuyue@fudan.edu.cn; mingda@mit.edu

It has been demonstrated that Weyl nodes may promote magnetic instabilities which lead to a helical magnetic order with an ordering wavevector matching the nesting of Weyl nodes[17,18], showcasing how magnetic properties can be engineered through band topology.

CeAlGe is an ideal platform to study the relationship between band topology and magnetism due to dual time-reversal and inversion symmetry breaking[19]. As a member of the non-centrosymmetric WSM RAlGe (R = La, Ce, Pr) family[19–22], CeAlGe shares the same body-centered tetragonal structure (space group $I4_1md$) with the proto-typical type-I WSM TaAs family[1,17,23] (Fig. 1a). Its magnetic structure has two inequivalent Ce sites, and prior studies have established its magnetic phases below $T_N$ = 4.5 K which lead to singular angular magnetoresistance[24] and the presence of an incommensurate topological magnetic meron-antimeron structure[25]. Given the intricate magnetic structure of CeAlGe, and the recent evidence for Weyl-mediated magnetism in isostructural NdAlSi and SmAlSi[17,18], it is natural to wonder if similar phenomena extend to this broader class of materials.

In this work, we find that Weyl fermions may help stabilize magnetic states in CeAlGe above its established magnetic transition temperature of $T_N$ = 4.5 K. By performing resonant elastic X-ray scattering (REXS), supported by a combination of electrical and thermal transport measurements, and dilatometry, we establish anomalous magnetic behavior in the temperature regime near $T$ = 12.8 K that has evaded prior scrutiny. In this regime, a number of unusual phenomena

emerge including peaked resonant diffraction intensity, zero magnetoresistance (MR), maximally suppressed magneto-thermal conductivity, and non-monotonic thermal expansion without any signature in magnetic susceptibility. We attribute the order to thermally-induced, topology stabilized nanoscale magnetic domains, which can be explained by our effective field theory. When the Weyl fermions couple with the magnetic domains with a finite-size effect considered, the interband scattering between Weyl points can dominate over the intraband scattering at certain wavevectors, resulting in a lowered free energy than the non-interacting case, and thereby stabilizes such order. Our experimental data meet the wavevector matching condition, in agreement with theory. Collectively, our results point to a manner in which the interaction between magnetism and topology can lead to order-by-disorder states which defy traditional thermodynamic intuition.

## Results

To explore the magnetic structure of CeAlGe, we carry out single crystal REXS measurements, which reveal a previously unreported nanoscale incommensurate magnetism above the thermodynamic magnetic transition temperature of $T_N$ = 4.5 K. Neutron powder diffraction from previous studies found that CeAlGe has a purely incommensurate ground state below $T_N$ = 4.5 K with a propagation vector $\mathbf{k}_{gs}$ = (0.066, 0.066, 0) before developing topological

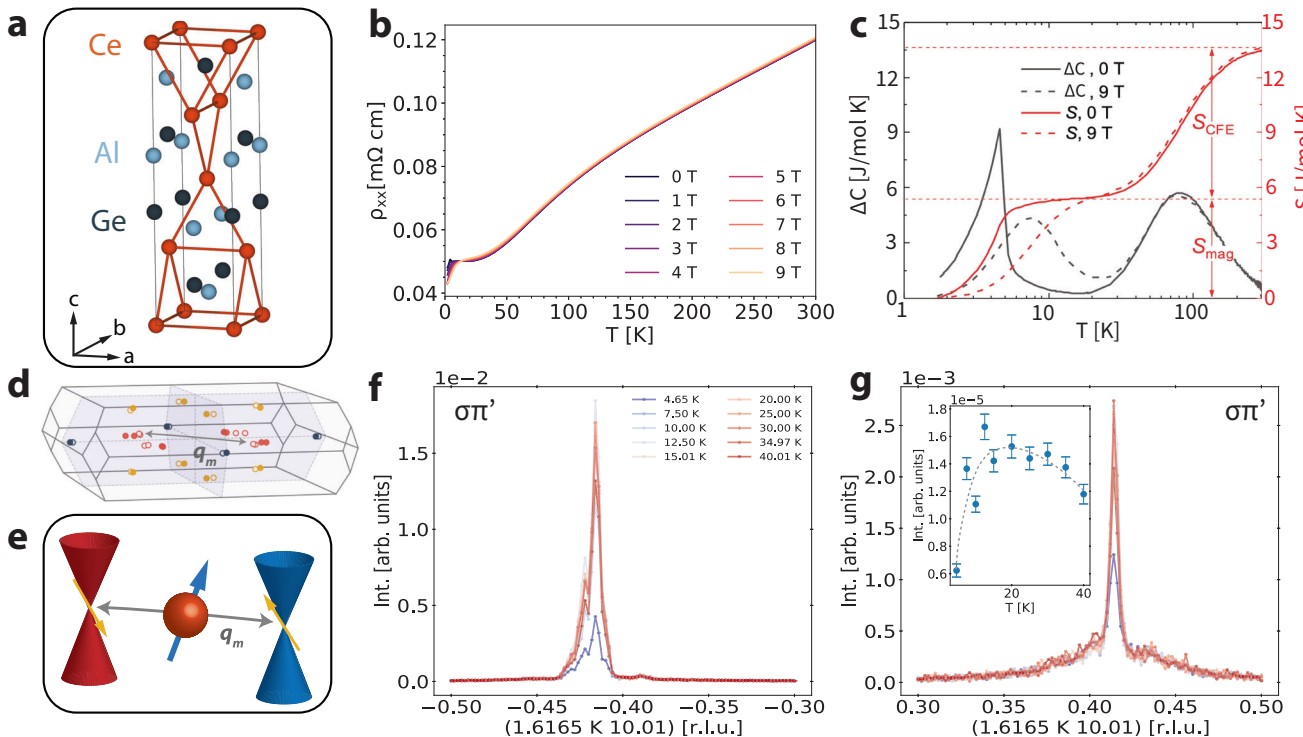

**Fig. 1 | Weyl-mediated incommensurate magnetism in CeAlGe. a** CeAlGe crystallizes in space group I4₁md (109) with broken inversion symmetry. Orange lines between the Ce atoms indicate the Ce-Ce bonds, highlighting the broken inversion symmetry. Below $T_N$ = 4.5 K, CeAlGe has a purely incommensurate magnetic ground state with ordering vector $\mathbf{k}_{gs}$ = (0.066, 0.066, 0)[25]. **b** Longitudinal resistivity from $T$ = 2 K to $T$ = 300 K at different magnetic fields from 0 to 9 T applied along the c-axis. **c** Difference of the heat capacities $\Delta C$ between CeAlGe and LaAlGe at magnetic fields of 0 T (black solid trace) and 9 T (black dashed trace). The entropy $S$ (right y-axis) is extracted through cumulative integration of the heat capacity $\Delta C$ at 0 T (red solid trace) and 9 T (red dashed trace). Major contributions are observed as resulting from crystal field splitting ($S_{CFE}$) at higher temperatures and the transition to the magnetic ground state at $T_N$ = 4.5 K ($S_{mag}$). **d** Location of the Weyl points within the Brillouin zone as obtained from ref. 19. $W_1$, $W_2$, and $W_3$ nodes are indicated in blue, yellow, and orange, respectively. The magnetic vector $\mathbf{q}_m$ is close to

the nesting condition between Weyl nodes. **e** Schematic of the nesting vector $\mathbf{q}_m$ between linearly-dispersive Weyl fermions. Red and blue Weyl cones have opposite chiralities, with yellow arrows indicating the spin degree of freedom for each cone. Orange atom depicts a localized magnetic ion whose spin (blue arrow) is coupled through indirect exchange to the spins of the itinerant Weyl electrons. **f, g** Elastic X-ray scattering resonant to the Ce L₂ edge in the σ-π′ channel shows an incommensurate magnetic ordering with wavevector $\mathbf{q}_m$ = (0.384, 0.416, 0) from a temperature range between $T$ = 4.65 K to $T$ = 40 K at both **f** negative and **g** positive K values in the (HKL) scattering plane. Trace temperature labels apply to each plot. The inset is a temperature dependence of the integrated incommensurate peak intensity at $\mathbf{q}_m$ = (0.384, ± 0.416, 0) normalized to the (0, 0, 8) structural Bragg peak. The dotted line serves as a visual guide. Error bars represent one standard deviation and are obtained through propagation of errors from counting statistics and Lorentzian peak fitting.

magnetism at finite magnetic fields applied along the $c$-axis[25]. The value of $T_N$ is consistent in our samples, as revealed through longitudinal electric resistivity (1b), heat capacity (Fig. 1c), and magnetic susceptibility (Supplementary Note 6) measurements under an applied magnetic field along the $c$-axis. REXS on the Ce $L_2$ edge (6.1642 keV) reveals a peak in the magnetic $\sigma$-$\pi'$ scattering channel with wavevector $\Delta\mathbf{k}$ = (1.6165, ±0.416, 10.01) (Fig. 1f, g) (Supplementary Note 9). We also find evidence of this incommensurate peak in elastic neutron scattering measurements, where the signal is averaged over many co-aligned single crystals (Supplementary Fig. 9 in Supplementary Note 7). This momentum transfer corresponds to a magnetic ordering with an incommensurate wavevector of $\mathbf{q}_m = \mathbf{Q}_{Bragg} - \Delta\mathbf{k}$ = (0.384, 0.416, 0). This wavevector coincides with that of possible nesting between the type-I $W_3$ Weyl nodes located at (±0.2, ±0.2, 0) shown in Fig. 1d and schematized in Fig. 1e (Supplementary Fig. 17 in Supplementary Note 10). Weyl-mediated incommensurate magnetism has been predicted[26–28] and observed in isostructural NdAlSi[17] and SmAlSi[18] arising from Dzyaloshinskii–Moriya interaction (DMI) and chiral terms that distinctly result from Ruderman-Kittel-Kasuya-Yosida (RKKY) interactions amongst the Weyl fermions. As shown in the inset of Fig. 1g, decreasing the temperature from $T = 40$ K changes the intensity of the magnetic ordering. The integrated intensity of the REXS peaks reaches a maximum near $T = 12.8$ K, where one does not observe an anomaly in magnetic susceptibility. This apparent inconsistency strongly suggests that the REXS signal is caused by short-ranged domains embedded within the paramagnetic background, whose orientation averages out to zero throughout the sample, and as indicated by the broad, diffuse background of the scattering signal in Fig. 1g. From the width of this diffuse background, we estimate the correlation length of the domains to be $\xi \approx 10$nm. As such, this ordering is within the temperature regime where thermal fluctuations dominate as indicated by the magnetic entropy saturation near $S_{mag} = R \ln 2 \approx 5.8$ J mol$^{-1}$ K$^{-1}$ (lower red dashed line in Fig. 1c) and explored further in electrical and thermal transport measurements.

The temperature dependence of the REXS peak is corroborated by transport and thermal expansion measurements, which show a distinct regime in temperature and magnetic field well above the magnetic transition temperature. Figure 2a and b show the second derivatives of longitudinal electrical resistivity $\rho_{xx}$ and thermal conductivity $\kappa_{xx}$, respectively, as a function of magnetic field and temperature (Supplementary Notes 2, 3, and 6). Identifying inflection points in transport properties is a powerful method for demarcating boundaries between distinct transport regimes in correlated systems[29], especially when the role of magnetism or strong interactions can induce a violation of the Wiedemann-Franz law. The data from thermal transport (Fig. 2b) and dilatometry (Fig. 2c) can be assigned to regions with thermal fluctuations, magnetic ordering, and field polarization. At temperatures below $T_N = 4.5$ K, by comparing with previous neutron diffraction reports[24,25], the low-temperature orderings can be assigned as the incommensurate (IC) ground state, the canted magnetic, and the fully-polarized ordering as the magnetic field increases. The electrical and thermal transport data at base temperature $T = 2$ K is plotted in Fig. 2e. In the ground state, electrical conductivity increases with increasing magnetic field due to the suppressed electron spin-flip scattering. Near $B = 3.3$ T, the Ce sites with smaller magnetic moment become fully aligned and the system enters the canted magnetic phase. Above $B = 6.2$ T, both Ce sites are field-polarized resulting in the fully-polarized ordering. In this region, the electrical conductivity decreases as field increases due to enhanced electron cyclotron orbital motion. On the other hand, thermal conductivity increases at higher field due to a larger magnon contribution. Above the magnetic transition, there are two regions which can be understood as the field-polarized regime and the spin fluctuation regime, resulting from the competition of thermal fluctuation and magnetic field alignment. The region dominated by thermal fluctuations is centered around $T = 10$–15 K at low field, as prominently featured through the thermal transport Fig. 2b and thermal expansion in Fig. 2c.

In the temperature region with thermal fluctuations near $T = 12.8$ K, we observe several anomalous features in the electric and thermal transport. At $T = 12.8$ K, the MR switches from being positive to negative for all measured values of applied magnetic field (Fig. 3a, d). Remarkably, at this exact temperature, the thermal conductivity is most suppressed by the external magnetic field (Fig. 3b). Such disparity between the electrical and thermal conductivities suggests that magnetic effects are the dominant contribution to the thermal transport. Similar MR behavior has been explored in other Ce-based intermetallics[30] where local nano-sized magnetic domains contribute to the resistivity even when there is no global magnetization, with their response governed by the interaction between conduction electrons and localized moments.

The physics of the interaction between local moments with itinerant charge carriers is also manifested through an upturn in the longitudinal resistivity as the temperature is decreased beginning near $T = 20$ K and zero applied magnetic field (Fig. 4). Such an upturn has been observed in rare-earth intermetallics in which moments from localized $f$-orbitals are screened by the conduction electrons, leading to a logarithmic temperature dependence of the resistivity[31]. More recent experiments have highlighted cases where an upturn is observed above the Néel temperature in materials with a magnetic ground state, implying the coexistence of Kondo exchange with magnetic ordering[32,33]. We fit the longitudinal resistivity data between $T = 15$ K and $T = 40$ K with a phenomenological model established by numerical renormalization group analysis of the Anderson impurity model with electron and phonon contributions[34,35]

$$R_{fit}(T) = R_0 + aT^2 + bT^5 + R_K(0)\left(\frac{T_K'^2}{T^2 + T_K'^2}\right)^s$$

where $T_K' = T_K/(2^{1/s} - 1)^{1/2}$ and $s = 0.225$ for a spin-1/2 local moment and $a$, $b$ are the coefficients of electron–electron scattering and electron–phonon scattering, respectively. The results of the fit are shown in Fig. 4 as red curves. The Kondo-like hybridization that leads to this upturn competes with the magnetic order which also arises from the local moment-itinerant electron interaction. Hence, when the nanoscale magnetic fluctuations begin to dominate the transport response below about $T = 15$ K, this phenomenological model does not fit the data anymore. Based on this resistivity upturn and the aforementioned transport behavior which indicates the presence of local magnetic order, we establish that the interaction between fluctuating local moments and itinerant conduction electrons of Weyl fermions is dominant in the temperature range near $T = 12.8$ K.

To model the interaction of the fluctuating local moments and the Weyl fermions, we consider the following effective Hamiltonian

$$H = H_0 + H_m + H_{int} \tag{1}$$

where $H_0$ is the non-interacting Hamiltonian of Weyl fermions, $H_m$ arises from the magnetization, and $H_{int}$ is the interaction of the Weyl fermions with the local magnetic moments. The free Weyl fermions Hamiltonian can be expressed as

$$H_0 = \sum_{i,\chi,\mathbf{k}}(\chi\nu_i v_F|\mathbf{k} - \mathbf{b}_i| + b_{0,i} - \mu)\psi_{i,\chi}^\dagger(\mathbf{k})\psi_{i,\chi}(\mathbf{k}) \tag{2}$$

where $i$ is the index for the Weyl points (WP) located at $(E, \mathbf{k}) = (b_{0,i}, \mathbf{b}_i)$, $\nu_i = \pm$ is the chirality charge of the Weyl fermions, $\chi = \pm$ is the index for alignment/anti-alignment spin-momentum locking, $\mu$ is the chemical potential, and $v_F$ is the Fermi velocity. Above the Néel

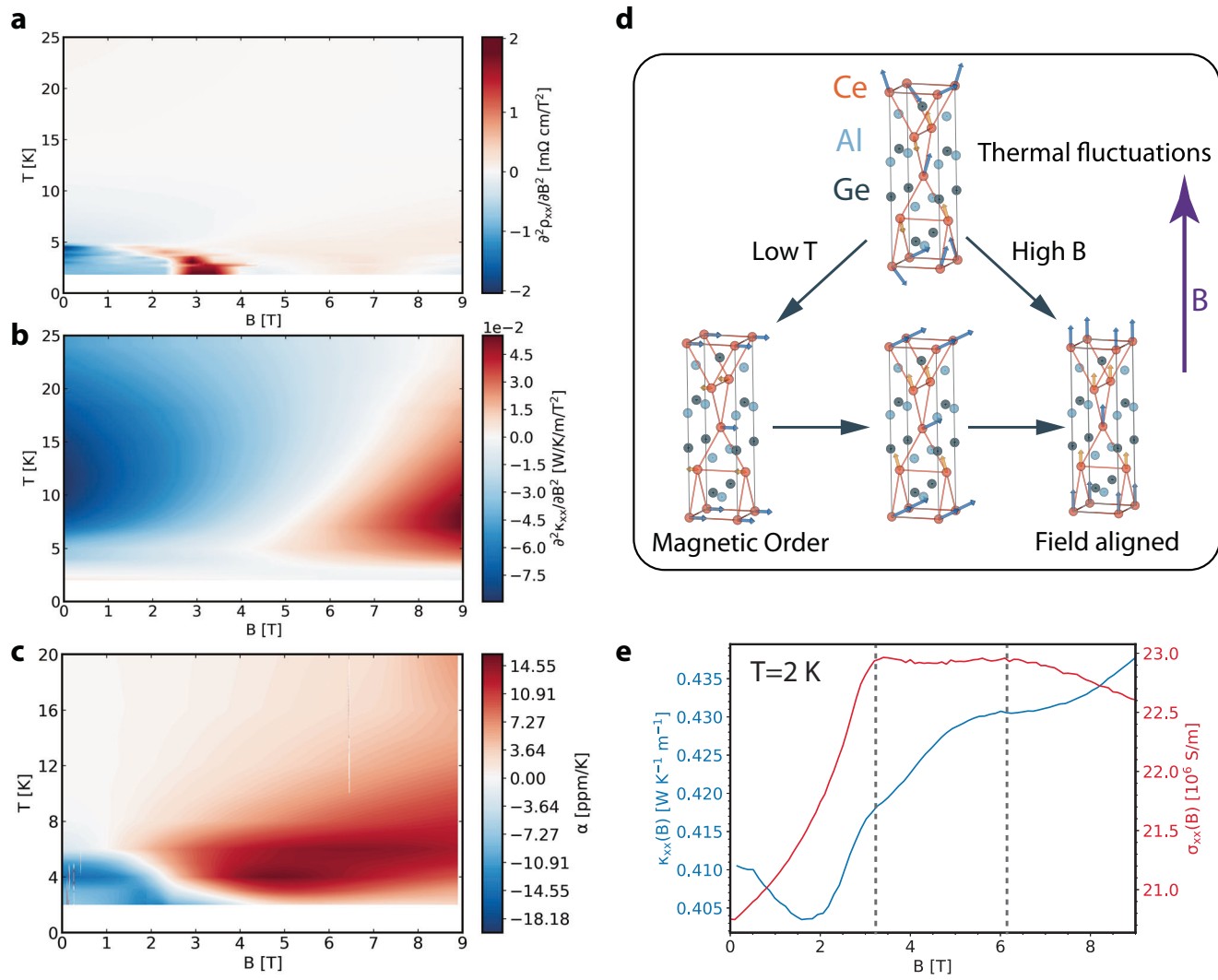

**Fig. 2 | Magnetic fluctuation regime revealed through transport and thermal expansion. a** Estimate of the phase diagram obtained from the second derivative of $\rho_{xx}$ with respect to $B$. Between 10 and 15 K and at low magnetic field, the longitudinal resistivity barely changes. **b** Estimate of the phase diagram obtained from the second derivative of $\kappa_{xx}$ with respect to $B$. Centered around 10–15 K, a distinct regime arises in the thermal transport at the same point in phase space where one observes anomalous electric transport. **c** Estimate of the phase diagram obtained from thermal expansion measurements with a high-resolution dilatometer, which also shows a distinct regime above thermodynamic transition temperature. Vertical streaks are artifacts from numerical interpolation. **d** Magnetic phases can be identified from transport behavior as regions where thermal fluctuations compete with intrinsic magnetic order and external field alignment. Blue and yellow arrows represent magnetic moments on the distinct Ce sites, and orange bonds between Ce atoms highlight the inversion symmetry breaking. Purple arrow indicates direction of the applied magnetic field. **e** Magnetic field-dependent behavior of the electric and thermal conductivities at $T = 2$ K follows the anticipated trend from magnetic structure configurations changing with external field. Vertical dashed lines demarcate distinct transport regimes.

temperature, we model the magnetic Hamiltonian as a paramagnet with DMI

$$H_m = \int d^3r \left( \frac{D}{4} \mathbf{M}(r) \cdot \nabla \times \mathbf{M}(r) + \frac{J}{2} (\nabla \mathbf{M}(r))^2 \right), \quad (3)$$

where $D$ is the DMI strength and $J$ is the exchange interaction strength. By only considering the regime of $T > T_N$, the complexity of the IC ground state can be avoided. This term gives rise to the chiral magnetization wave upon minimizing the Hamiltonian[36] as

$$\mathbf{M} = M(\hat{e}_1 \cos(\mathbf{q}_m \cdot r) + \hat{e}_2 \sin(\mathbf{q}_m \cdot r)), \quad (4)$$

where $\hat{e}_1 \times \hat{e}_2 = \hat{e}_3 \equiv \mathbf{q}_m/|\mathbf{q}_m|$ and $|\mathbf{q}_m|$ is defined from $D$ and $J$.

We introduce the Kondo interaction between the local moment and the Weyl fermions with the following interaction Hamiltonian

$$H_{int} = \int d^3r \left( \sum_i K_{\nu_i} \mathbf{M}(r) \cdot s_{\nu_i}(r) \right) \quad (5)$$

where $s_\nu(r) = \frac{1}{2} \psi_\nu(r)^\dagger \sigma \psi_\nu(r)$ is the single-particle spin operator and $K$ is the Kondo coupling strength between the local moment and itinerant Weyl fermions. The Eq. (5) can be rewritten as

$$H_{int} = \sum_{\mathbf{k},\nu,\alpha,\beta} \frac{K_\nu M}{2} \left( \psi_{\nu\alpha}^\dagger(\mathbf{k}) V_{\alpha\beta} \psi_{\nu,\beta}(\mathbf{k} + \mathbf{q}) + \text{h.c.} \right) \quad (6)$$

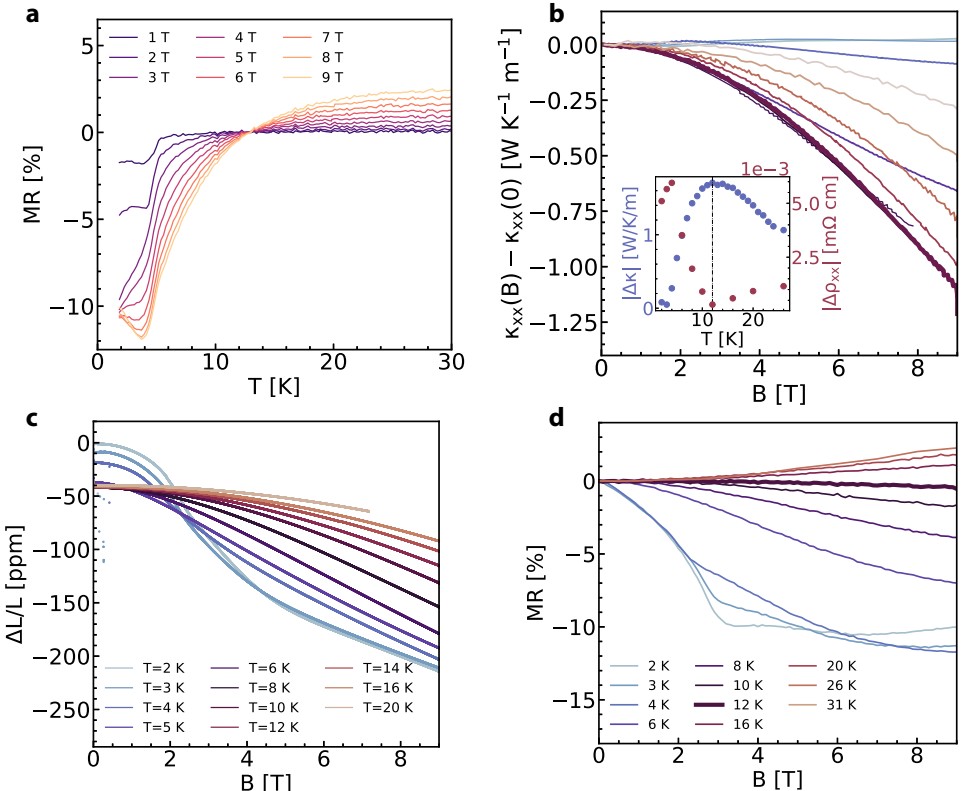

**Fig. 3 | Anomalous transport at $T = 12.8$ K with B‖c. a** The MR = $100*[\rho(B, T) - \rho(B = 0, T)]/\rho(B = 0, T)$ exhibits a crossover with temperature from positive to negative values around $T = 12.8$ K. **b** At this temperature, the thermal conductivity is greatly suppressed with increasing magnetic field. The inset shows comparison of maximum change in thermal conductivity and electrical resistivity. The dashed line demarcates $T = 12$ K which is the point at which the difference with and without magnetic field in the thermal conductivity is maximal and that in the resistivity is minimal. Legend is shared with panel **d**, **c**. Low temperature dilation measurements. **d** The MR plotted as a function of applied magnetic field demonstrates the close-to-zero value at $T = 12.8$ K.

with

$$V_{++} = e^{i\phi} \cos(\theta'/2) \sin(\theta/2)$$
$$V_{--} = e^{i\phi'} \cos(\theta'/2) \sin(\theta/2)$$
$$V_{+-} = \cos(\theta/2) \cos(\theta'/2)$$
$$V_{-+} = -e^{i\phi} e^{i\phi'} \sin(\theta/2) \sin(\theta'/2).$$

In this form, the interaction potential can be interpreted as $\chi$-preserving scattering ($V_{++}, V_{--}$) and $\chi$-flipping scattering ($V_{+-}, V_{-+}$) between two WPs with identical chiral charge, denoted in Fig. 5a.

By considering the scattering event between two WPs with the same chiral charge and a momentum space separation $\Delta\mathbf{b}$, we can perturbatively calculate the free energy contribution at $\mu = b_0$. If we consider a contribution from a pair of WPs with identical chiral charge and renormalize the momentum with respect to the Fermi energy, we can express the second-order free energy contribution as (Supplementary Note 12)

$$F^{(2)} = \left(\frac{KM}{2}\right)^2 \sum_{\mathbf{k}} \left[ \operatorname{Re}\left[V_{++}\bar{V}_{++}\right] \frac{n(|\bar{\mathbf{k}}|) + n(-|\bar{\mathbf{k}}+\mathbf{q}|)) - n(-|\bar{\mathbf{k}}|) - n(|\bar{\mathbf{k}}+\mathbf{q}|)}{|\mathbf{k}+\mathbf{q}| - |\mathbf{k}|} \right.$$
$$\left. - \operatorname{Re}\left[V_{+-}\bar{V}_{-+}\right] \frac{n(-|\bar{\mathbf{k}}|) + n(-|\bar{\mathbf{k}}+\mathbf{q}|)) - n(-|\bar{\mathbf{k}}|) - n(|\bar{\mathbf{k}}+\mathbf{q}|)}{|\mathbf{k}+\mathbf{q}| + |\mathbf{k}|} \right] \quad (7)$$

where

$$\tilde{\mathbf{k}} = v_F \mathbf{k} = \tilde{k}(\sin\theta\cos\phi, \sin\theta\sin\phi\cos\theta)$$
$$\mathbf{q} = v_F(\mathbf{q_m} - \Delta\mathbf{b})$$
$$\tilde{\mathbf{k}} + \mathbf{q} = |\tilde{\mathbf{k}}+\mathbf{q}|(\sin\theta'\cos\phi', \sin\theta'\sin\phi', \cos\theta')$$

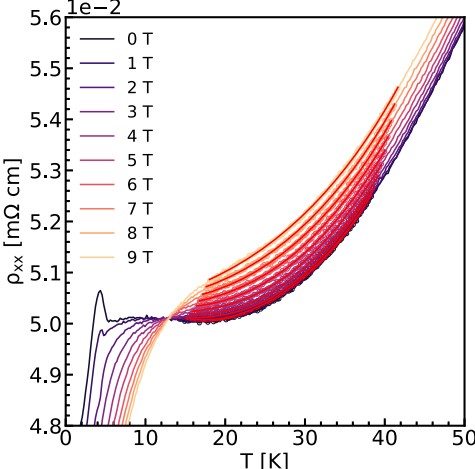

**Fig. 4 | Resistivity minimum near $T = 20$ K.** Further evidence for conduction electron interaction with local moments is manifested through a resistivity upturn and minimum near 20 K. Longitudinal resistivity data are fitted as a function of field and temperature with the theoretical behavior of the Kondo model. Fits are shown in red solid lines.

is the renormalized momentum and $n(E)$ is the Fermi-Dirac distribution function (details in Supplementary Information).

Eq. (7) is the central result and shows how the coupling between Weyl fermions and magnetic fluctuations could stabilize the magnetic order. Physically, the contributions correspond to conduction-conduction/valence-valence (intraband) scattering $\operatorname{Re}[V_{++}\bar{V}_{++}]$ and

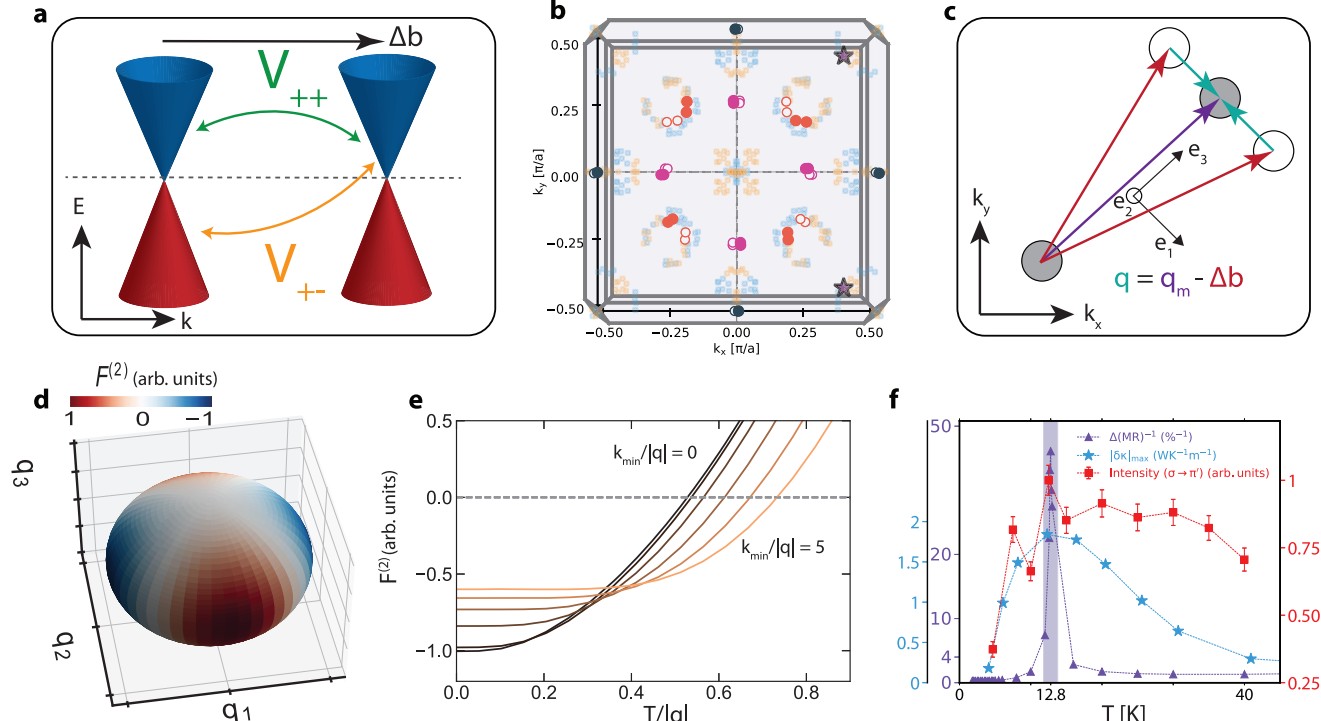

**Fig. 5 | Local order stabilization. a** Scattering between two Weyl nodes with the same chirality, separated in momentum space by **Δb**. The dashed line denote the Fermi level of the bands. Two kinds of scattering are possible, intraband scattering $V_{++}$ and interband scattering $V_{+-}$. **b** Nesting conditions between Weyl nodes. Large circles represent the positions of the Weyl nodes within the $k_z = 0$ plane (adapted from[19]), with open and closed circles indicating nodes of different chiralities. Smaller circles represent all possible momentum space separations **Δb** of the Weyl nodes, with blue and orange representing scattering between nodes of the same and opposite chiralities, respectively. Purple stars represent the location of the measured magnetic wavevector, **$q_m$**. **c** Components of scattering between two Weyl nodes of the same chirality **q** (green arrows) based on the Weyl node separation **Δb** (red arrows) and magnetic wavevector **$q_m$** (purple arrow). Note that the $k_x - k_y$ coordinate is material coordinate while the $e_1 - e_2 - e_3$ coordinate is the magnetic wavevector coordinate described in Eq. (4). **d** The free energy contribution from the interaction $F^{(2)}$ and **q** in $q_1 - q_2 - q_3$ coordinate basis of the magnetic propagation vector. The contribution is negative when the scattering wavevector is perpendicular to the propagation direction of the magnetization, lowering the free energy. **e** $F^{(2)}$ as a function of dimensionless temperature $T/|\mathbf{q}|$ for different domain size cutoffs where dimensionless $k_{min}/q$ is inversely proportional to the domain size. As the domains decrease in size, the free energy is negative over a larger temperature range. **f** Summary of the inverse of total MR deviation, maximal field-induced thermal conductivity deviation, and REXS data as a function of temperature in the fluctuation-stabilization regime. Error bars representing one standard deviation for the first two quantities are smaller than the marker size. The uncertainty in the REXS data is calculated from proper error propagation of the Poissonian statistics in the number of detector counts and error related to the fitting of the incommensurate peak via a Lorentzian with a background function. The dashed lines serve as a visual guide of the trend and the purple shaded region denotes the temperature region near 12.8 K.

conduction-valence (interband) scattering $Re[V_{+-}\bar{V}_{-+}]$. While the intraband scattering always contributes positively $Re[V_{++}\bar{V}_{++}]>0$, the interband scattering can have a negative sign depending on the angle of the scattering, as shown in Fig. 5c, d. The negative free energy contribution can be interpreted as stabilizing the local moment from the scattering process, which flips spin to align with the local moment.

At $T = 0$, the intraband scattering and the free energy contribution as a function of **q** alignment are shown in Fig. 5d. We note that when the scattering momentum **q** is aligned purely with the propagation direction of the magnetization wavevector $\hat{e}_3$, the free energy contribution is zero as the scattering $\mathbf{q}_m$ preserves the spin component of the electron in the plane of the chiral magnetization. On the other hand, when the scattering momentum **q** has a non-zero component perpendicular to the propagation direction and is in the magnetization plane, (along $\hat{e}_1 - \hat{e}_2$ directions), spin scattering couples with the local moment and contributes to free energy. The sign of this contribution depends on whether the scattered spin has lower or higher magnetic energy than the initial spin.

With multiple WPs, the total free energy is the sum contribution from every pair of Weyl-node scattering (including self-scattering). However, since the free energy scales with $1/|(\tilde{\mathbf{k}}+\mathbf{q}|+|\tilde{\mathbf{k}}|) \sim 1/|\mathbf{q}|$ at low temperature, only the free energy contribution from the WP-pairs

that are almost commensurate with $\mathbf{q}_m$ play a role in determining the stability of the local moment. In addition, we have shown that the free energy is lowest when $q_1$, is maximal. The observed $\mathbf{q}_m$ can be explained as the result of this consideration where $\mathbf{q}_m$ is aligned such that it is almost commensurate with the WP nesting and $q_1$ is non-zero, as set by the nesting condition from another set of Weyl nodes.

At finite temperatures, the positive contribution to the free energy from intraband scattering (first term in Eq. (7)) competes with the interband contribution and wins over at high enough temperatures (Fig. 5e). Above the magnetic transition temperature, while the magnetic ordering is globally unstable, certain magnetic moments could be locally stabilized as a domain. To understand the impact of a finite size domain in the free energy calculation, we introduce a low energy cutoff for the sum over momentum. $\sum_{\tilde{\mathbf{k}}} \mapsto \sum_{|\tilde{\mathbf{k}}| > k_{min}}$ where $k_{min} \sim 1/L$ depends on the size of the domain $L$. As the cutoff $k_{min}$ is increased, the intraband scattering contribution decays significantly faster than the interband contribution, and the negative free energy persists to higher temperatures, as shown in Fig. 5e. At low enough temperature, however, the system undergoes the full magnetic transition and the the nano-sized domains compete with the ground state order. In this way, these domains are only stable in a narrow temperature window just

above the magnetic transition temperature before dying off at higher temperatures.

## Discussion

Our model highlights that the distinct attributes of Weyl-nodes–namely their topological protection–are crucial in stabilizing local magnetic fluctuations, regardless of the origin of the helical magnetization (Weyl-nesting or not). In materials where the helical direction is not set by the Weyl-node nesting in the first place, our proposed mechanism still applies, but the strength of the coupling is likely to be weak, since there is no guarantee that the magnetic wavevector $\mathbf{q}_m$ is close to Weyl node separation $\mathbf{\Delta b}$. In isostructural materials NdAlSi[17,37] and SmAlSi[18] incommensurate magnetic order is found to be nearly commensurate with Weyl-node nesting, which theory predicts to contribute to large $D$ interaction strength despite a small density of states at the Fermi surface. However, in materials with multiple Weyl nodes, it has not been clear why one magnetic direction may be favored over others given how many Weyl node nesting conditions there can be. By considering additional scattering events between Weyl nodes, our model reveals the conditions under which the helical magnetization arising from Weyl node nesting is stable in some directions more than in others. Specifically, when $\mathbf{q}_m$ is close to another nesting condition between two Weyl nodes $\mathbf{\Delta b}$ we expect the fluctuation stabilization to be most pronounced.

Our observations in several experimental probes (REXS, electric and thermal transport, and dilatometry) collectively point towards the presence of nanoscale magnetic domains above the thermodynamic magnetic transition temperature. We develop a theory which proposes that these magnetic fluctuations are stabilized by the coupling between Weyl fermions and local magnetic moments in CeAlGe. Other recent experiments in geometrically frustrated materials have also highlighted the prominence of magnetic fluctuations and their impact on transport[14,38]. While these studies focus on the role of lattice geometry in fluctuation stabilization, the model we develop highlights the role of k-space topology. The development of time-resolved X-ray scattering measurements such as X-ray photon correlation spectroscopy[39,40] may help further elucidate the nature of these fluctuations by observing their temporal dynamics. Given the crucial role that magnetic fluctuations play in correlated quantum materials, such as in unconventional superconductors[41] and light-driven order[42], we anticipate broader responses to arise from the interactions between magnetic fluctuations and electronic band topology.

## Methods

### Synthesis of CeAlGe single crystals and powder

We synthesized high-quality single crystals of CeAlGe through the Al self-flux method. A mixture of Ce powder (Strem Chemicals, 99.9%), Al beads (Sigma-Aldrich, 99.9%) and Ge powder (Beantown Chemical, 99.999%) were weighed in a molar ratio of 1:10:1 in a glovebox and placed into a crucible. The mixture-filled crucible was flame-sealed in an evacuated quartz tube and was subsequently heated up to 1100 °C from room temperature at a rate of 80 °C/h. Afterwards, the mixture dwelled for 20 h and subsequently cooled to 700 °C at a rate of 3 °C/h. This was followed by several days of annealing at this temperature after which centrifugation was performed to remove the excess flux. The resulting products of CeAlGe single crystals approximately half-centimeter large and have a metallic luster with lattice constants $a = 4.29$ Å and $c = 14.74$ Å as measured with powder X-ray diffraction. A total of 10 g of CeAlGe in powder form were also prepared via a solid-state reaction for the time-of-flight neutron scattering experiments. The Ce, Al and Ge powders were weighed in a 1:1:1 molar ratio and placed in a crucible which was flame-sealed in an evacuated quartz tube. The materials were calcined at 700 °C. The resulting products

were ground and flame-sealed in quartz tube to be annealed at 700 °C for several days.

### Electrical and thermal transport

To perform electrical and thermal transport experiments, the single crystals of RAlGe (R = La, Ce) were thinned along the $c$ axis to a thickness of 0.265 mm and 0.370 mm, respectively. The magneto-transport measurements (longitudinal resistivity and thermal conductivity) were acquired using a Physical Property Measurement System (PPMS) Dynacool (Quantum Design) using a diagonal offset probe geometry. A more detailed description of these measurements, along with the data processing, can be found in the Supplementary Information.

### Dilatometry

Sample dilation was measured with an ultrasensitive differential capacitive dilatometer produced by Quantum Design[43] inserted into a PPMS Dynacool. At each temperature, data was taken at both positive and negative magnetic field and the resulting dilatometry was symmetrized.

### Resonant elastic X-ray scattering

High-precision hard X-ray scattering measurements were performed at Beamline 4-ID of the National Synchrotron Light Source II (NSLS-II) at Brookhaven National Laboratory. A photon energy of 6.164 keV, corresponding to the Ce $L_2$ edge, is selected using a cryogenically-cooled Si(111) double-crystal monochromator. The photon energy was varied between values slightly below (6.124 keV) and slightly above (6.204 keV) the resonance peak to monitor the energy dependence. Further details are described in the Supplementary Information.

### Inelastic neutron scattering (INS)

Inelastic time-of-flight neutron scattering experiments were carried out on the fine-resolution Fermi chopper spectrometer SEQUOIA[44,45] at the Spallation Neutron Source (SNS) in Oak Ridge National Laboratory (ORNL). We utilized co-aligned single crystals of CeAlGe on an aluminum plate using X-ray diffraction with a total mass of 2.8 g. Data were collected with different incident energies $E_i = 4$ meV, 12 meV, 30 meV, 60 meV, 240 meV, 1000 meV at base temperature (1.5 K) and at 10 K which is above $T_N$. The runs at $E_i = 4$ meV and 12 meV were performed on the high resolution setting of the instruments whereas the other incident energies were carried out on the high flux setting enabling measurements downwards to 2% and 5% energy resolution, respectively. An external magnetic field was applied along the $c$-axis of the CeAlGe single crystals with measurements performed at 0 T, 1 T, 3 T, and 8 T.

Inelastic triple-axis neutron scattering experiments were carried out on the polarized triple-axis spectrometer PTAX (HB-1) and on the triple-axis spectrometer TAX (HB-3) at the High Flux Isotope Reactor in Oak Ridge National Laboratory (ORNL). In these triple-axis experiments, we also used co-aligned CeAlGe single crystals with a total mass of 4 g. We used a fixed $E_f = 14.7$ meV with 48′-40′-40′-120′ collimation and Pyrolytic Graphite filters to eliminate higher-harmonic neutrons. Measurements were performed using helium cryomagnet refrigerators at base temperature (1.5 K) and at a temperature of 10 K in tandem with applied magnetic fields along the $a$- and $c$-axis of the single crystals of 0 T, 1 T, 3 T, 5 T, and 8 T. Measurements of the spin waves described in the main text were performed along (h10), (h02), and (hh0). The bulk of the measurements concentrated on the optical phonon dispersion of CeAlGe along (h00) at an energy range between 10 meV and 25 meV.

## Data availability

The data used in this study are available in the Figshare database under accession code https://doi.org/10.6084/m9.figshare.23576637.

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

## Acknowledgements

NCD, TN and FH acknowledge the support from U.S. Department of Energy (DOE), Office of Science (SC), Basic Energy Sciences (BES), Award No. DE-SC0020148. NCD, TN and ZC acknowledge National Science Foundation (NSF) Designing Materials to Revolutionize and Engineer our Future (DMREF) Program with Award No. DMR-2118448. TL and ML are partially supported by NSF Convergence Accelerator Award No. 2235945. ML acknowledges the support from Class of 1947 Career Development Professor Chair. SH acknowledges support from National Science Foundation (NSF), Award No. 2230400 and Welch Foundation Award No. C-2144. The research on neutron scattering used resources at Oak Ridge National Laboratory's High Flux Isotope Reactor (HFIR) and Spallation Neutron Source (SNS) which are sponsored by the Scientific User Facilities Division, Office of Basic Energy Sciences, U.S. Department of Energy. Raman measurements were conducted at the Center for Nanophase Materials Sciences, which is a DOE Office of Science User Facility. The X-ray scattering measurements used resources of the National Synchrotron Light Source II, a U.S. Department of Energy (DOE) Office of Science User Facility operated for the DOE Office of Science by Brookhaven National Laboratory under Contract No. DE-SC0012704 and of the Advanced Photon Source, a U.S. Department of Energy (DOE)

Office of Science User Facility operated for the DOE Office of Science by Argonne National Laboratory under Contract No. DE-AC02-06CH11357.

## Author contributions

ML conceived and supervised the project. NCD, TN, and FH performed the transport measurements with the support from ML. FH, TN, and QTN synthesized the materials. TN, NCD, FH, NA, ML performed neutron scattering measurements with help from TJW, MBS, AIK, SC, JFB. NCD, TN, ZC, and LKN performed magnetometry. TN, NCD, FH, NA, ML performed X-ray scattering measurements with help from CSN and AA. TN, FH, QTN, AAP performed Raman measurements with support from DBG and SH. PS, XL, YY and GB developed theory. ZZ performed the ab initio calculations. NCD, TN, PS and ML wrote the paper with input from all authors.

## Competing interests

The authors declare no competing interests.
