## [Peer Review File · Nature Communications]

REVIEWER COMMENTS

Reviewer #1 (Remarks to the Author):

The authors report the evidence for thermally induced short-range magnetism. There are two main findings: 1. The wave vector of the order matches with the nesting of Weyl nodes. 2. There is transport anomaly at 12.8K, at which the strongest intensity of the short-range order is also observed. Here are some comments/questions:

1. The fact that the wave vector matches with the nesting of Weyl nodes is interesting. However, since similar effects are already reported in Nat. Mater. 20, 1650 (2021) (as the authors point out), the novelty of the work is compromised.
2. Can the authors specify the magnetic field direction in figure 3(for MR)?
3. If the $T_c=12.8K$ anomaly is related to the formation of magnetic domains, why is it independent of the magnetic field (in figure 3a)?
4. It is not very clear to me how the interactions of local moment and itinerant electron cause the anomaly presented in figure 3. It will be better if the authors can provide more discussions.

Overall, I think the findings in the work are interesting, but the origin/underlying relation of those findings is still elusive to me.

Reviewer #2 (Remarks to the Author):

This manuscript by Drucker et al., focuses on the emergence of an anomalous magnetic phase at 12.8 K above the magnetically ordered state, which is stabilized by Weyl fermions. This magnetic region is manifested by a number of unusual phenomena like peaked resonant diffraction intensity, crossover from positive to negative value in MR, suppressed thermal conductivity and non-monotonous dilation. All these observations point towards the presence of nano-scale magnetic domains. However, there are several concerns that need to be addressed before recommending it, for publication.

- 1) Different regions (I to V) are assigned in the figure 2 (b). However, both III and IV regions are described by field-polarized regions. How these regions are different from each other and what is the origin of demarcating these regions is not clear.
- 2) The origin of nano-scale magnetic domains is not reflected in the REXS analysis part of the manuscript. A detailed discussion regarding this is required for a clearer picture.
- 3) The physical meanings of the parameters (p, q,) in the given equation (in page 5) are not explained.
- 4) How Kondo temperature is related to the observed magnetic state at 12.8 K? Is it signifying the presence of any kind of impurity/crystal defect in the studied sample? A clarification regarding this is required in the manuscript.
- 5) Also, in the figure 4, fitting is done in the region 15 – 40 K. Then how the obtained Kondo temperature is related to the observed feature at 12.8 K?
- 6) Figure 5 (b) is given in the manuscript, but is not discussed in the text.
- 7) The conclusion of the manuscript needs to be modified as it is not reflecting the whole manuscript.

REVIEWER COMMENTS

Reviewer #1 (Remarks to the Author):

The authors report the evidence for thermally induced short-range magnetism. There are two main findings: 1. The wave vector of the order matches with the nesting of Weyl nodes. 2. There is transport anomaly at 12.8K, at which the strongest intensity of the short-range order is also observed. Here are some comments/questions:

1. The fact that the wave vector matches with the nesting of Weyl nodes is interesting. However, since similar effects are already reported in Nat. Mater. 20, 1650 (2021) (as the authors point out), the novelty of the work is compromised.
2. Can the authors specify the magnetic field direction in figure 3(for MR)?
3. If the $T_c=12.8K$ anomaly is related to the formation of magnetic domains, why is it independent of the magnetic field (in figure 3a)?
4. It is not very clear to me how the interactions of local moment and itinerant electron cause the anomaly presented in figure 3. It will be better if the authors can provide more discussions.

Overall, I think the findings in the work are interesting, but the origin/underlying relation of those findings is still elusive to me.

We thank the reviewer for taking the time to read the manuscript and provide this constructive feedback. We have revised the manuscript substantially to address these comments directly in the following way. In particular, we are pleased to share that we have constructed a new comprehensive effective field theory, we are able to depict the physical mechanism undergoing in this system, and to explain under what conditions such topology stabilized order can emerge.

From our theoretical calculations, we show that there are three criteria for such local magnetic ordering to emerge in the classical disorder regime ($T \gg T_N$):

- a) Presence of Weyl nodes
- b) The Weyl nodes separation is close enough to the magnetic nesting vector
- c) Finite size effect of the magnetic domain is taken into account

All three criteria a) – c) are met and observed in experiments. Under these conditions, the inter-band Weyl fermion scattering will dominate over the intra-band scattering, resulting in a lowered free energy of the interacting system and thus explains the observed ordering, as schematically Shown in the new Fig. 5a,c and quantitatively calculated in Fig. 5d and 5e.

In addition to the comprehensive theory, below please find our point-by-point response to each questions rendered.

1. The fact that the wave vector matches with the nesting of Weyl nodes is interesting. However, since similar effects are already reported in Nat. Mater. 20, 1650 (2021) (as the authors point out), the novelty of the work is compromised.

As the reviewer points out, we have observed a similar phenomena – Weyl node nested magnetism—that others have recently been exploring. Our work builds on these other studies in the following way:

- a. The reports in NdAlSi and SmAlSi both only report this behavior alongside an associated 2nd order phase transition. **We observe this behavior *above* the phase transition to the ground state magnetic order, which begs the question of why it is still stable locally.**
- b. In our updated version of the manuscript, we develop a theoretical model which considers local moment-Weyl fermion coupling in an extension of previous studies. In particular, we explore the role of helical magnetization—Weyl fermion coupling in the free energy of the magnetization regardless of the origin of that magnetization (from Weyl node nesting or some other mechanism). We find that this stabilization effect is most enhanced for the case when the magnetic nesting is close in momentum to the scattering between Weyl-nodes, which is likely to be the case if the magnetism also arises from Weyl-node nesting. Nevertheless, we emphasize that our results are not limited to systems with helical magnetism originating from Weyl-node nesting.

2. Can the authors specify the magnetic field direction in figure 3(for MR)?

The magnetic field direction in all figures is along the c-axis of the crystal unit cell. The caption of the figure has been updated for clarity.

3. If the T_c=12.8K anomaly is related to the formation of magnetic domains, why is it independent of the magnetic field (in figure 3a)?

The anomaly in MR at T=12.8K is independent of magnetic field because of the competition between nonmagnetic *and* magnetic contributions to the magnetoresistance. As discussed in Sluchanko et al. Low Temp. Phys. 41, 1011 (2015) and established by K. Yosida, Phys. Rev.107, 396(1957), negative magnetoresistance (nMR) has contributions from local magnetization (M_{loc}) $-\frac{\Delta\rho}{\rho} \propto M_{loc}^2$ while conventional positive magnetoresistance arising from cyclotron orbits is positive and proportional to square of applied field $\frac{\Delta\rho}{\rho} \propto B^2$. When magnetic field is applied, the local magnetization is proportional to the magnetic field, and it is possible for these two contributions, one negative and one positive, to balance out at a particular temperature. In this way, T=12.8K is the crossover temperature when the local magnetization becomes strong enough to balance out the conventional cyclotron contribution to MR. Below this temperature, the magnetic fluctuations of the helical magnetic order are weaker (as observed in REXS), but there are also fluctuations of the ground state magnetic order which compete with both the helical magnetization and the conventional MR contribution. Thus, we see nMR even above the magnetic transition temperature.

4. It is not very clear to me how the interactions of local moment and itinerant electron cause the anomaly presented in figure 3. It will be better if the authors can provide more discussions.

In Figure 3, we primarily present transport data which includes hallmarks of magnetic behavior, as discussed in the main text and addressed in the prior comment. When magnetism is present, it affects the transport of itinerant electrons as established by K. Yosida, Phys. Rev. 107, 396(1957). Notably, even above the magnetic transition temperature, in this model of the s-d exchange interaction of local moments and itinerant electrons, local magnetization can still impact transport properties.

Another key piece of evidence for the local moment- itinerant electron interaction is presented in Figure 4, in which the longitudinal resistivity exhibits an upturn *before* the magnetic transition at low temperature. This resistivity upturn is a hallmark of the Kondo interaction between local moment and itinerant electrons, as explored beginning in the 1960s and discussed thoroughly in P. Coleman, in Handbook of Magnetism and Advanced Magnetic Materials (2015). Notably Doniach showed [Doniach, S. (1977). Phase Diagram for the Kondo Lattice. In: Parks, R.D. (eds) Valence Instabilities and Related Narrow-Band Phenomena. Springer, Boston, MA. <https://doi.org/10.1007>] that in a lattice of local moments, the interaction between itinerant fermions and local moments has two distinct regimes. The resulting Doniach phase diagram scales with the magnetic exchange strength and the electron density of states. In one regime of the Doniach phase diagram, the RKKY magnetic exchange interaction dominates and the material has a magnetic ground state. In the other regime of the phase diagram, the Kondo interaction wins out and a heavy fermion ground state is formed by local moments screened by the itinerant fermions.

Although these two regimes are thermodynamically distinct, recent experiments in Ce-based intermetallics have highlighted the fact that these two effects can compete in the same material. In experiments, this competition manifests as an upturn in resistivity *before* the magnetic ground state transition, implying that while the kondo interaction may be present, the magnetic interaction is stronger and thus forms the ground state [S. Jang, et al. , Sci. Adv. 5, eaat7158 (2019), C. E. Matt, et al. , Phys. Rev. B 105, 085134 (2022)]. In our data, the upturn can be modeled using the model established by renormalization analysis of the Anderson impurity model [T. A. Costi, A. C. Hewson, and V. Zlatic, J. Phys. Condens. Matter 6, 2519 (1994)] in addition to contributions from electron-electron scattering and electron-phonon scattering. As shown in Figure 4, this model fits the data very well (red curves).

As shown in the figure on the left [Sato, Y. (2021). Introduction. In: Quantum Oscillations and Charge-Neutral Fermions in Topological Kondo Insulator YbB₁₂. Springer Theses], T_k is lower than T_N in the limit of low J and magnetism is the ground state of the Kondo lattice because it is energetically favored. In our data, the extracted Kondo temperature is very low (60mK) which can be explained by the fact that the upturn is very small in our data, suggesting the presence of some Kondo screening despite the magnetic order. Nevertheless, since the ground state of the system is magnetic and *not* a heavy fermion, we have removed this specific mention of T_k since its interpretation is not straightforward, other than the qualitative statement that there is some

degree of Kondo screening. This piece of evidence helps justify the use of local moment-itinerant electron coupling in the Hamiltonian explored in the theory section.

Reviewer #2 (Remarks to the Author):

This manuscript by Drucker et al., focuses on the emergence of an anomalous magnetic phase at 12.8 K above the magnetically ordered state, which is stabilized by Weyl fermions. This magnetic region is manifested by a number of unusual phenomena like peaked resonant diffraction intensity, crossover from positive to negative value in MR, suppressed thermal conductivity and non-monotonous dilation. All these observations point towards the presence of nano-scale magnetic domains. However, there are several concerns that need to be addressed before recommending it, for publication.

We thank the reviewer for their careful reading of the paper and constructive criticism. Below, we comment on each of the points raised by the reviewer, and highlight the changes we have made in the updated manuscript. In particular, we are pleased to share that we have constructed a new comprehensive effective field theory, we are able to depict the physical mechanism undergoing in this system, and to explain under what conditions such topology stabilized order can emerge.

From our theoretical calculations, we show that there are three criteria for such local magnetic ordering to emerge in the classical disorder regime ($T \gg T_N$):

- a) Presence of Weyl nodes
- b) The Weyl nodes separation is close enough to the magnetic nesting vector
- c) Finite size effect of the magnetic domain is taken into account

All three criteria a) – c) are met and observed in experiments. Under these conditions, the inter-band Weyl fermion scattering will dominate over the intra-band scattering, resulting in a lowered

free energy of the interacting system and thus explains the observed ordering, as schematically Shown in the new Fig. 5a,c and quantitatively calculated in Fig. 5d and 5e.

In addition to the comprehensive theory, below please find our point-by-point response to each questions rendered.

1) Different regions (I to V) are assigned in the figure 2 (b). However, both III and IV regions are described by field-polarized regions. How these regions are different from each other and what is the origin of demarcating these regions is not clear.

In the original manuscript, these regions were demarcated by points where the second derivative of thermal conductivity with respect to field are zero. This method has been used to demarcate phase boundaries in correlated materials, where thermal conductivity takes into account quasiparticle transport beyond electrical conductivity, which only includes contributions from charged quasiparticles. **However, upon reviewing the manuscript and listening to feedback from the reviewers, we recognize that these demarcations are more confusing than helpful.** Thus, we have removed them from the manuscript.

2) The origin of nano-scale magnetic domains is not reflected in the REXS analysis part of the manuscript. A detailed discussion regarding this is required for a clearer picture.

We have updated the section of the paper where we discuss the diffuse background of the REXS data in fig. 1g (reproduced below).

In the scattering data at the magnetic peak (left and blue on right), there is a significant broad, diffuse background, which indicates the presence of short-range correlations. Notably, this background is broader than the structural Bragg peak (orange in left, which is superimposed for the sake of argument). We can fit a Lorentzian (green in right figure) to the diffuse background and extract the correlation length, which is on the order of 10nm.

3) The physical meanings of the parameters (p, q,) in the given equation (in page 5) are not explained.

These parameters represent the coefficients of the electron-electron and electron-phonon scattering contributions to the low temperature resistivity data. We have updated the manuscript accordingly.

4) How Kondo temperature is related to the observed magnetic state at 12.8 K? Is it signifying the presence of any kind of impurity/crystal defect in the studied sample? A clarification regarding this is required in the manuscript.

We address this question and the next one below.

5) Also, in the figure 4, fitting is done in the region 15 – 40 K. Then how the obtained Kondo temperature is related to the observed feature at 12.8 K?

A key piece of evidence for the local moment- itinerant electron interaction is presented in Figure 4, in which the longitudinal resistivity exhibits an upturn *before* the magnetic transition at low temperature. This resistivity upturn is a hallmark of the Kondo interaction between local moment and itinerant electrons, as explored beginning in the 1960s and discussed thoroughly in P. Coleman, in Handbook of Magnetism and Advanced Magnetic Materials (2015). Notably Doniach showed [Doniach, S. (1977). Phase Diagram for the Kondo Lattice. In: Parks, R.D. (eds) Valence Instabilities and Related Narrow-Band Phenomena. Springer, Boston, MA. <https://doi.org/10.1007>] that in a lattice of local moments, the interaction between itinerant fermions and local moments has two distinct regimes. The resulting Doniach phase diagram scales with the magnetic exchange strength and the electron density of states. In one regime of the Doniach phase diagram, the RKKY magnetic exchange interaction dominates and the material has a magnetic ground state. In the other regime of the phase diagram, the Kondo interaction wins out and a heavy fermion ground state is formed by local moments screened by the itinerant fermions.

Although these two regimes are thermodynamically distinct, recent experiments in Ce-based intermetallics have highlighted the fact that these two effects can compete in the same material. In experiments, this competition manifests as an upturn in resistivity *before* the magnetic ground state transition, implying that while the kondo interaction may be present, the magnetic interaction is stronger and thus forms the ground state [S. Jang, et al. , Sci. Adv. 5, eaat7158 (2019), C. E. Matt, et al. , Phys. Rev. B 105, 085134 (2022)]. In our data, the upturn can be modeled using the model established by renormalization analysis of the Anderson impurity model [T. A. Costi, A. C. Hewson, and V. Zlatic, J. Phys. Condens. Matter 6, 2519 (1994)] in addition to contributions from electron-electron scattering and electron-phonon scattering. As shown in Figure 4, this model fits the data very well (red curves).

As shown in the figure on the left [Sato, Y. (2021). Introduction. In: Quantum Oscillations and Charge-Neutral Fermions in Topological Kondo Insulator YbB₁₂. Springer Theses], T_k is lower than T_N in the limit of low J and magnetism is the ground state of the Kondo lattice because it is energetically favored. In our data, the extracted Kondo temperature is very low (60mK) which can be explained by the fact that the upturn is very small in our data, suggesting the presence of some Kondo screening despite the magnetic order. Nevertheless, since the ground state of the system is magnetic and *not* a heavy fermion, we have removed this specific mention of T_k since its interpretation is not straightforward, other than the qualitative statement that there is some

degree of Kondo screening. This piece of evidence helps justify the use of local moment-itinerant electron coupling in the Hamiltonian explored in the theory section.

6) Figure 5 (b) is given in the manuscript, but is not discussed in the text.

We have an updated figure 5 which reflects the theory section of the paper. We have still included the experimental data part (now figure 5f) but have removed the schematic of local moment-Weyl fermion coupling.

7) The conclusion of the manuscript needs to be modified as it is not reflecting the whole manuscript.

We have updated the conclusion of the manuscript to include discussion of the theoretical model of Weyl-stabilized magnetic fluctuations in relation to the experimental data. We also highlight other recent work on lattice-stabilized fluctuations in topological materials [K. K. Kolincio, et al, Proc. Natl. Acad. Sci. U.S.A 118, e2023588118 (2021), K. K. Kolincio, et al. , Phys. Rev. Lett. 130, 136701 (2023)] in contrast with our k-space mechanism. Finally, we highlight the role of fluctuations in light-induced phases of matter, a field of research which is gaining momentum.

REVIEWER COMMENTS

Reviewer #1 (Remarks to the Author):

In the updated version, the authors have provided a comprehensive theory to explain the observations. I think the manuscript has been improved substantially and I believe the paper may be published in its present form.

Reviewer #2 (Remarks to the Author):

In the revised manuscript, the authors have responded well to most comments and have done the required corrections. The new comprehensive effective field theory have significantly improved the manuscript. Hence, I recommend the manuscript for publication.

Reviewer #3 (Remarks to the Author):

This paper presents an experimental study of a claimed magnetic Weyl semimetal material CeAlGe above its magnetic transition temperature.

Short-range correlated magnetically-ordered states are observed and are claimed to be evidence of "topology-stabilized" magnetic order.

While I think this work is not without some merit and should be published somewhere, I do not think it needs to be published in a high-profile

journal such as Nature Communications. A few more specific comments are below.

1. I believe the claim of "Weyl-mediated magnetism" to be strongly overstated and misleading (this also applies to Ref. 17 in my opinion).

The presence of several nested Fermi surfaces does not require Weyl nodes, the fact that they enclose them is an accident.

DM interactions, that the authors invoke, also do not require Weyl nodes, only broken inversion symmetry.

2. The theoretical analysis that the authors use is far from convincing. The authors seem to rely on the property that the band Hamiltonian

near the Weyl nodes has the form $H = \pm \sigma \cdot k$, where σ is the spin that couples to the local magnetic moments. This is way oversimplified

in my opinion. While σ is certainly related to spin, it will certainly contain admixture of the orbital degrees of freedom and how exactly it couples

to the local moments is complicated, nonuniversal, and may only be deduced from a careful analysis of the electronic structure of a given material.

I doubt the theoretical analysis presented in this paper has much to do with the reality.

In summary, I do not recommend this paper to be published in Nature Communications. I would suggest this be submitted to a more specialized journal.

Reviewer #3 (Remarks to the Author):

This paper presents an experimental study of a claimed magnetic Weyl semimetal material CeAlGe above its magnetic transition temperature.

Short-range correlated magnetically-ordered states are observed and are claimed to be evidence of "topology-stabilized" magnetic order.

While I think this work is not without some merit and should be published somewhere, I do not think it needs to be published in a high-profile journal such as Nature Communications. A few more specific comments are below.

1. I believe the claim of "Weyl-mediated magnetism" to be strongly overstated and misleading (this also applies to Ref. 17 in my opinion).

The presence of several nested Fermi surfaces does not require Weyl nodes, the fact that they enclose them is an accident.

DM interactions, that the authors invoke, also do not require Weyl nodes, only broken inversion symmetry.

2. The theoretical analysis that the authors use is far from convincing. The authors seem to rely on the property that the band Hamiltonian

near the Weyl nodes has the form $H = \pm \sigma \cdot k$, where σ is the spin that couples to the local magnetic moments. This is way oversimplified

in my opinion. While σ is certainly related to spin, it will certainly contain admixture of the orbital degrees of freedom and how exactly it couples

to the local moments is complicated, nonuniversal, and may only be deduced from a careful analysis of the electronic structure of a given material.

I doubt the theoretical analysis presented in this paper has much to do with the reality.

In summary, I do not recommend this paper to be published in Nature Communications. I would suggest this be submitted to a more specialized journal.

We thank the Reviewer for their careful reading of the paper and constructive criticism. In the following, we respectfully address the specific comments.

1) I believe the claim of "Weyl-mediated magnetism" to be strongly overstated and misleading (this also applies to Ref. 17 in my opinion).

The presence of several nested Fermi surfaces does not require Weyl nodes, the fact that they enclose them is an accident. DM interactions, that the authors invoke, also do not require Weyl nodes, only broken inversion symmetry.

There are several significant reasons to believe that the simultaneous presence of Weyl-nodes and helical magnetism is not an accident.

First, the criterion for Weyl-mediated magnetism is stringent, it is located at very well-defined Q -points, set by the location of the Weyl-nodes. In fact, *our work also takes into account the case where the magnetic wavevector is close to the Weyl-nodes but not necessarily induced by them.* Nevertheless, the observation of helimagnetism very near Weyl-node nesting in a growing family of materials suggests that these results are **not ‘accidental’ but corroborate each other, precisely the reason to systematically understand the interplay between electronic Weyl-nodes and magnetic degrees of freedom.**

Second, while the DM interaction does not require Weyl-nodes, it is caused by the carriers near the fermi level through an antisymmetric exchange interaction. This interaction is dependent on the density of states at the fermi-level. However, the unique nature of the Weyl-nodes (spin-momentum locking) creates a sizeable DM interaction despite a small electron DOS at the fermi level for the topological bands (17, 18). Even though the DM does not require Weyl-nodes in the most basic case, in CeAlGe--along with other isostructural materials NdAlSi and SmAlSi-- the carriers at the fermi-level are Weyl-nodes, and we argue that it is necessary to include their contribution to the DM interaction. Ultimately, we do not believe that our conclusion is in contradiction with the statement that **DM interactions, that the authors invoke, also do not require Weyl nodes, only broken inversion symmetry.**

2) The theoretical analysis that the authors use is far from convincing. The authors seem to rely on the property that the band Hamiltonian near the Weyl nodes has the form $H = \gamma \mathbf{p} \cdot \boldsymbol{\sigma}$, where $\boldsymbol{\sigma}$ is the spin that couples to the local magnetic moments. This is way oversimplified in my opinion. While $\boldsymbol{\sigma}$ is certainly related to spin, it will certainly contain admixture of the orbital degrees of freedom and how exactly it couples to the local moments is complicated, nonuniversal, and may only be deduced from a careful analysis of the electronic structure of a given material. I doubt the theoretical analysis presented in this paper has much to do with the reality

- a) In any real material, a minimum model is adopted to avoid introducing unnecessary parameters while also being abstractable to other systems. Despite the relative simplicity of the model we introduce relative to the complexity of the ground state in CeAlGe $T < T_N$ (several incommensurate magnetic orders, including a topological mero-antimeron structure, and the only known material with two incommensurate magnetic propagation vectors, see Figure below), we believe that **our experimental evidence lends credence to the use of specific terms, especially the $H = \gamma \mathbf{p} \cdot \boldsymbol{\sigma}$ term.** This is the term corresponding to the Weyl-dispersion that has chiral spin momentum locking, and calculations show that these contribute to the fermi level. In addition, we use this model to explore the physics in the simpler regime at $T > T_N$, which takes away most of the complexities associated with the ground state. It is also compelling to use this term to describe the interaction between local moments that have helimagnetic structure at wavevector Q which matches the calculated Weyl-node separation (accidental or not, they are very close). While other parts of the fermi surface may contribute to such scattering, it is

those parts closest to Q —the Weyl nodes—which will contribute the most, and thus using this term in the Hamiltonian is appropriate to describe the scattering process.

INCOMMENSURATE STRUCTURES

One propagation vector

1.1.1 Cs_2CuCl_4

1.1.2 $\text{RbFe}(\text{MoO}_4)_2$

1.1.3 Cr

1.1.4 Cr

1.1.5 CaFe_4As_3

Click to expand/compact...

Two propagation vectors

2.1.1 CeAlGe

- b) Even considering the orbital degree of freedom in CeAlGe (e.g. DFT calculations in PRB 97, 041104), it only led to a simple (but wrong) ferromagnetic ordering, without given the correct AFM order and disagrees with any experimental observations. Therefore the necessity to add orbital degrees of freedom is unjustified and the DFT calculation already indicates the minimum role orbital degrees of freedom play.

We believe that our experimental findings, backed up by a minimal model that is not over complicated, corroborate with and expand upon previous studies of Weyl-mediated magnetism (17,18 in our manuscript). Importantly, our theory points to the way in which Weyl-nodes may stabilize helimagnetic fluctuations, *even when the helimagnetism is not caused by the Weyl-nodes*. Thus we feel that the reviewer's comments are taken into account already in the manuscript.

REVIEWERS' COMMENTS

Reviewer #3 (Remarks to the Author):

The authors have responded to my comments in a satisfactory manner.

As far as I am concerned, the paper may be published in present form.